# The Synergy of Thermal and Non-Thermal Effects in Hyperthermic Oncology

**DOI:** 10.3390/cancers16233908

**Published:** 2024-11-21

**Authors:** Carrie Anne Minnaar, Gyula Peter Szigeti, Andras Szasz

**Affiliations:** 1Department of Radiation Sciences, University of the Witwatersrand, Johannesburg 2000, South Africa; 2John von Neumann Faculty of Informatics, Óbuda University, 1034 Budapest, Hungary; 3MedTech Innovation and Education Center, University Research and Innovation Center, Óbuda University, 1034 Budapest, Hungary; 4Department of Biotechnics, Hungarian University of Agriculture and Life Sciences, 2100 Gödöllő, Hungary

**Keywords:** heterogenic heating, cellular selection, thermal processes, non-thermal actions, immunogenic cell death (ICD), damage-associated molecular pattern (DAMP), abscopal effect, cancer vaccination

## Abstract

Modulated electro-hyperthermia combines the thermal and non-thermal effects of an applied electromagnetic field. This synergy allows for the selection of malignant cells with a minimal load on healthy cells. The modulated radiofrequency induces immunogenic effects, which promote the targeting of malignant cells in the system by activating the circulating tumour-specific killer cells, thereby acting as a vaccination against the tumour.

## 1. Introduction

The therapeutic use of heat has a long history in medicine, epitomized by the saying, “Give me the power to produce fever, and I will cure all diseases” [1]. Historically, the aim was to achieve the highest possible temperatures for successful treatment outcomes and tumour destruction. However, without understanding the physiological and molecular effects of heat, these objectives were unrealistic. Despite advances in heating techniques, modern hyperthermia (HT) practices still rely on these early goals. Classical HT involves maintaining isothermal conditions by heating specific regions, using temperature as the sole measure of clinical efficiency.

Hyperthermia has not yet achieved widespread recognition in oncology, and even those who are actively using HT have not reached a consensus regarding its feasibility. Despite successful clinical trials [2,3,4] and high-level, comprehensive meta-analyses [5,6,7,8] providing convincing results, HT’s acceptance is suboptimal. This may be due to the challenges associated with the related physics [9] or physiology [10]. These obstacles can be overcome with more knowledge and research into the field [11].

To cause the desired substantial cellular damage to malignant tissue required in classical local–regional hyperthermia (LRH), temperatures above 41–42 °C are needed [12]. In practice, the varied depth of tumours and the regulation of thermal homeostasis limits the actual temperatures achieved to 39–42 °C [13]. The technical differences within LRH techniques may result in differences in survival benefits, as can be seen when comparing separate studies on cervical carcinoma treated with radiative [14] and capacitive [15] LRH methods. A further challenge in LRH is that the non-homogeneous heating of the tumour complicates the CEM43 dose measurement (cumulative equivalent minutes at 43 °C [CEM43] [16]). To solve this dosing challenge, the area (%) that achieves the desired temperature is also considered [17]. Despite the challenges, several trials have validated oncologic HT as a curative and palliative therapy.

Electromagnetic technologies using radiofrequency (RF) and microwaves (MW) dominate the field of HT. Capacitive-coupled heating techniques use RF technology and radiative heating devices use MW technology. Historically, the goal of heating was to cause necrosis, which was shown to occur in vitro at 43 °C [18]. Both heating technologies have challenges in achieving the desired temperature safely without causing burns in the surrounding healthy tissues or the adipose layers. Magnetic Resonance (MR)-guided thermometry technology has been developed to detect and monitor hot spot formation to reduce the risk of damage to healthy tissues; however, this adds considerable cost and complexity to the treatment [19]. The selection and depth of the tumour pose additional challenges; however, these can be partly corrected by applying multiple energy sources around the body and focusing the energy absorption at the tumour location—a technique employed by rHT [20]. Targeting heat-energy-absorbing particles within the tumour provides another potential heating method that would spare the healthy tissues that do not have the heat-energy-absorbing particles. In this method, the nanoparticles are artificially inserted into the tumour [21] and targeted, heating up the malignant tissue. Methods of flushing these nanoparticles from the body after the treatment are still being investigated.

An alternate option to the nano-heating concept would be to target naturally occurring molecular groups found in higher concentrations in tumours with a particular ability to absorb more energy than other molecules [22]. The membrane lipid raft structures, which appear in higher numbers in malignant cells, have thermal stability and appear to be promising targets for heating [23]. The simulation by Papp et al. showed that the membrane rafts were excited, and the heat increased in the presence of a 13.56 MHz RF-generated electromagnetic field. In the simulation, the current density reached 23,000 A/m^2^. The specific absorption rate (SAR) in the rafts was high, and the energy loss was therefore minimal [24].

Modulated electro-hyperthermia (mEHT) is a capacitive heating technique that employs the use of amplitude modulation of the RF wave at a frequency of 1/f. The modulation has non-thermal effects that promote the selection and apoptosis of malignant cells [25,26]. Another important difference between mEHT and classical cHT techniques, which also apply 13.56 MHz RF waves, is the precise impedance matching, which minimizes energy loss and maximizes SAR, allowing for a lower power output and improved safety. The resulting electric field has a current density low enough to excite the molecules (non-thermal excitation) without being overpowered by the resulting thermal energy production [27]. Preclinical studies on mEHT show that mEHT can trigger cell death signals, which significantly improve cellular degradation synergistically with heat. These molecular effects are triggered by the non-thermal excitation of the chemical bonds. Non-thermal excitation, which is promoted by the amplitude modulation, is not seen in classical cHT techniques [28]. The balanced synergy of the thermal and non-thermal processes appears to optimize the antitumoural effect of mEHT and has shown to promote systemic homeostatic controls such as immune surveillance for metastatic disease [29].

This review summarises the molecular and physiological heating processes achieved by mEHT, with a focus on the synergy of the thermal and non-thermal effects of electromagnetic energy absorption.

## 2. Thermal Load

Conventional HT centres around the thermal effects and measures its success with temperature, which is a parameter of the thermal processes. The various thermal effects induce a complex set of regulatory mechanisms through homeostatic surveillance. The thermal load causes multiple reactions in the tumour, which changes various parameters associated with the tumour’s interaction with the tumour microenvironment (TME). The wide range of molecular changes impacts the tumour’s local conditions, and these changes cannot be described by measuring the temperature alone. Non-thermal components accompany the thermal effects [30], modifying the temperature-dependent thermal processes [31].

The thermal processes modify the chemical reaction rate (k) of a reaction with $E_{a}$ activation energy. The Arrhenius law (1) empirically describes the linear dependence of ln(k) on the reciprocal of the absolute temperature (1/T) [32,33]:(1)$$k={Ae}^{-\frac{E_{a}}{R_{G}T}}\;\;\;⟹\;\ln\left(k\right)=\ln\left(A\right)-\frac{E_{a}}{R_{G}}\frac{1}{T}$$
where $R_{G}\approx 8.3\;\frac{J}{mol\cdot K}$ is the universal gas constant. The cell membrane may undergo a phase transition when the lipid structure transforms from one state to another [34], which appears at a breaking point on the straight line of a logarithmic plot [35]. The reaction rate influences the parameters of the Arrhenius plot, with a substantial reduction (almost ⅓) in $E_{a}$ above this point [36]. The activation energy $E_{a}$ and pre-exponential factor A vary, but the linear dependence of (1) remains valid [37].

Usually, we define the thermal processes by the existence of the (1) Arrhenius plot. The break point of the plot serves as a fixed point for the HT dose. The measured break point is $T_{break}=42.5\;{}^{\circ}C$ in vitro [16] and so it is used in the standardisation of HT [17], recognizing necrosis at $T_{necrotic}=43\;{}^{\circ}C$.

Heat absorption and its thermal effects differ at microscopic structures that are driven by the heterogeneous electrolyte distribution, which is separated by lipid membranes. The following primary micro-absorptions modify the TME and cytosolic structures [38]:Due to the different thermal and electrical properties, the temperature differs in the extracellular matrix (ECM) and cytosol. The cell membrane is not conductive when exposed to frequencies below 15 MHz due to its capacitor-like properties, which result in minimal current flow across it at these lower frequencies. This behaviour changes at higher frequencies, where the membrane becomes more conductive, allowing greater current flow and reducing the selective difference between current flow through the extracellular matrix (ECM) and the cytosol. The membrane’s insulating properties dominate at frequencies below 15 MHz, maintaining distinct current pathways in the ECM and cytosol during RF heating processes. Furthermore, different current flows exist in the ECM than the cytosol, which is isolated by the membrane during RF heating processes. When exposed to lower frequencies (<1 kHz), the cell membrane acts as a capacitor due to its lipid bilayer, which can impede the flow of the electric current. The impedance of the membrane is relatively high, which means that low-frequency fields have limited penetration and effects on the cell [39]. As the frequency increases, the impedance of the cell membrane decreases. This is because the capacitive reactance of the lipid bilayer decreases with increasing frequency. At frequencies between 1 kHz and 100 kHz, the membrane’s resistance to the flow of the electric current is reduced, allowing for greater conductivity. The interaction with the fields at frequencies higher than 100 kHz becomes more complex and for frequencies in the MHz range, other phenomena, such as the skin effect, also come into play. The membrane becomes almost conductive at f>15 MHz, and so at high frequencies, this type of selection difference between the current flow through the ECM and cytosol fades. The current flow heats the electrolytes differently on the two sides of the membrane during low-frequency (f<15 MHz) heating. Due to its high metabolic rate, the TME conducts the current better than the ECM in the healthy host tissue; so, the temperature difference (temperature gradient) between TME and the cytosol of malignant cells is higher.The heat conductivity of the membrane wall allows heat to flow from the ECM to the cytosol, forced by the temperature gradient between the two sides of the membrane [40]. However, the rapid temperature equalization throughout the membrane does not mean that the mass of the cytosol reaches the same temperature as the TME due to the larger volume and specific heat capacity of the cytosol. Studies of cellular heat responses indicate that temperature variations across cellular structures can lead to localized thermal effects, without necessarily equalizing the temperature throughout the entire cytosolic mass immediately [41]. Heating up the mass of cytosol requires a high amount of energy coming through the membrane, which takes a longer time. This is supported by the general principles of thermodynamics.

Malignant cells are highly mobile and can form temporary cell-to-cell contacts [42]. The inhomogeneous electric field appears to trigger intensive dielectrophoretic interactions in the 0.1–100 MHz intervallum [43]. The dielectric permittivity (ε) of the transmembrane proteins is at least two orders of magnitude higher than the membrane permittivity in which they move [44], and so the dielectrophoretic force acts selectively on them. They produce high a SAR at the contact points, causing an increase in temperature of the transmembrane protein clusters (rafts). The mechanical contacts of the cells can also absorb high amounts of energy [24].

## 3. Non-Thermal Load

The chemical processes can also be non-thermal, involving interactions that heat cannot achieve [45]. The thermal state stabilizes the condition of the chemical machinery of living objects. The Arrhenius law (1) describes thermally induced chemical reactions. A subtle thermal effect that results in an unmeasurable temperature increase is also non-thermal when the temperature measurements are not accurate or sensitive enough. Intensive non-thermal characteristics appear where the thermal environment provides optimal conditions. The non-thermal molecular excitations point to the biomatter’s natural heterogeneities [46].

The thermal effects promote chemical changes, and the modified chemical structures may cause a change in the temperature development, as appears in the Arrhenius plot’s kink. Additional to the thermal changes, the electric field non-thermally modifies some chemical structures and actively manipulates the biomatter with electromagnetic interactions, like polarization, excitation, percolation, aggregation, electrophoresis, electroporation, electro-osmosis, etc. The electric field affects numerous bioprocesses, such as endocytosis [47], regulating and controlling biosynthesis [48], acting on the articular cartilage [49], playing a role in the treatment of inflammation [50], and pain management [51]. The external field may excite the transmembrane proteins, inducing signals transferred through the complex multi-component intracellular pathways. The non-thermal stimuli also impact intracellular compartments like the mitochondria [52], ER [53], Golgi apparatus, the nuclei, and the cytoskeletal network [54].

Many essential bioprocesses follow non-thermal kinetics. During a phase transition, the structure and chemical bonds of cells are rearranged [55], and although this is a thermal process, the temperature does not change during the transition. The quantum mechanics and resonances in chemical bonds are mostly non-equilibrium thermodynamic states demonstrating the synergy of thermal and non-thermal energy components [56]. The external electric field non-thermally promotes cellular fission at low [57] and high [58] frequencies, and it has promising applications in oncology [59].

### 3.1. Cellular Membrane

The cell membrane’s electric charge (membrane potential) attracts ions from the surrounding matrix, which are essential for cell signalling [60]. These ions can form bonds with the membrane [61], altering its non-thermal functions and lowering the influence of temperature-related activities. These create a significant electrical potential difference across the membrane, which can impact how the cell operates [62].

### 3.2. Cancer

Non-thermal charge transfer happens at the micro and macro levels. The movement of the ions impacts the development of cancer by regulating the apoptosis–proliferation balance [63]. The collagen network in the ECM also affects the transport of the large molecules and their carried charges [64]. The non-thermal electrokinetic and electro-osmotic processes have the potential to move fluids and particles in tissues. It is therefore possible that these processes can help manage issues that are problematic for traditional heat-based treatments, such as fluid build-up from ascites, pleural effusion, and oedema [65].

Electric fields can be used to control cellular behaviour non-thermally [66]. The induced current determines the orientation [67] and the dynamics of cell division [65], and it promotes cellular migration to heal wounds [68]. Malignant cells are more negative in their charge on their surface than their healthy counterparts, and their membrane potential is markedly lower [69]. This results in a potential gradient between the tumour and its healthy surroundings [70], which produces an electric current. As occurs during the wound healing process, the generated electric current promotes and directs cell migration [71] and supports the dissemination of the malignant cells. In this way, the malignant tissue appears to mimic a wound, demanding support from the healthy host tissues [72] and behaving as if it is a chronic injury [73]. The permanent current stimulates the proliferation of malignant cells. Feng et al. propose that this behaviour may contribute to the tumour’s ability to avoid detection by immune surveillance [74].

## 4. The Thermal Condition for a Non-Thermal Load

The pre-exponential factor $\left(A\right)$ and the activation energy $\left(E_{a}\right)$ in Equation (1) proportionally change in the biosystems [75,76] as follows:(2)$$E_{a}=a\cdot \ln\left(A\right)+b$$
where a and b are constants $a≅2.6\cdot 10^{3}$ and $b≅2.5\cdot 10^{4}$ [75] or $b≅2.6\cdot 10^{4}$ [75]. The strict correlation of $E_{a}$ and A shows that the thermal effects are in synergy with non-thermal components in biological media. The reaction rate is calculated from Equations (1) and (2) as follows:(3)$$\ln\left(k\right)=\ln\left(A\right)-\left(\frac{a\cdot ln\left(A\right)+b}{R}\right)\frac{1}{T}$$

So, the slope of the Arrhenius plot becomes $s_{Arrh}^{\prime }=-\left(\frac{a\cdot ln\left(A\right)+b}{R}\right)$, instead of $s_{Arrh}=-\left(\frac{E_{a}}{R}\right)$. $E_{a}$ depends on ln(A), and so the dynamism of the processes (represented in A) modifies the slope. As a result, the heating speed substantially modifies the heating effects [77], the technique used influences the slope [78], and the kink position varies depending on the species in which the measurements are taken [17]. The significant differences in the biological effects of homogeneously water-bath-heated (wHT) and heterogeneously heated mEHT at the same temperature [28] indicate the complex dynamic mechanisms involved in each technique.

The empirical Arrhenius law has a chemically adequate quantum mechanical approach in which the Arrhenius law is modified [79,80], and is expressed in the Eyring formula [81]:(4)$$K=\kappa \frac{k_{B}T}{h}e^{\frac{-\Delta G^{‡}}{RT}}=\kappa \frac{k_{B}T}{h}e^{\frac{\Delta S^{‡}}{R}}e^{\frac{-\Delta H^{‡}}{RT}}$$
where κ is the reaction coefficient, h is the Plank constant ($h≅6.63\cdot 10^{-34}\;J\cdot s$), $k_{B}≅1.38\cdot 10^{-23}\left[\frac{J}{K}\right]$ is the Bolzman constant, and $\Delta G^{‡}$, $\Delta S^{‡}$ and $\Delta H^{‡}$ are the transition Gibbs energy, transition entropy, and transition enthalpy, respectively. In this approach, the pre-exponential factor $\left(A\prime \right)$ characterizes the transition state:(5)$$A\prime =\kappa \frac{k_{B}T}{h}e^{\frac{\Delta S^{‡}}{R}}$$

The temperature remains constant during the first-order phase transition, while the pre-exponential factor changes according to the transition entropy $\Delta S^{‡}$. The transition enthalpy (the energy in chemical systems, like bonds, solvation, etc.) depends on the final configuration but does not vary according to the process used to achieve it. The transition entropy $\Delta S^{‡}$ refers to the unavailability of thermal energy to convert into mechanical work, which is understood as the degree of randomness or disorder in the system. These developing quantum mechanical principles [82] have impacts on a variety of applications.

The electric field can facilitate reactions or transitions by providing energy in a non-thermal manner, but with effects comparable to heating [18]. The mEHT optimizes the thermal/non-thermal synergy [83]. The external electric field non-thermally excites the transmembrane proteins using the thermal conditions to produce apoptotic signals for cell death [28]. The non-thermal component decreases the activation energy in the Arrhenius barrier (Appendix A). The selection at the cellular level is non-thermal. The electromagnetic heterogeneity of the target determines the radiofrequency signal path and thus automatically selects the highly conductive and highly dielectrically permeable microenvironment of cancer cells.

## 5. Heating Selection of Malignant Tissue Using Inherent Tissue Heterogeneities

The inherent heterogeneity in malignant tissue structures, driven by differences in cancer cell size, shape, and metastatic potential, affects energy absorption. Malignant tissue can be recognized through its varying cell morphology, abnormal intercellular network, as well as dynamic properties, such as heat conduction, convection, and radiation. The variations in conductivity and dielectric permittivity can significantly change how electrical and magnetic processes occur in the tissue [84].

Due to the heterogenic complexity of malignant tissue, the same power flow [$W/m^{2}$] may cause different energy absorptions depending on the given conditions [85], the organ [86], and the frequency [87].

### 5.1. Macro Selection

The increased metabolic activity of the malignant cells alters the TME, resulting in increased glucose metabolism (Warburg effect [88]) and a higher ionic concentration in the TME, which creates a more conductive path for the electric current. As a result, the tumour has higher conductivity (σ) than its healthy surroundings [89,90]. Furthermore, the higher water content of the malignant lesion [91] makes the ECM electrolytes more conductive [30], increasing the tumour’s conductivity and enhancing the selection of the malignant tissue by the electric field.

The disorder in the malignant tissue, resulting from the broken intercellular bonds and junctions, allows for the easier polarization of molecules, increasing the tumour’s electric permittivity (*ε*) and decreasing the electric impedance to further facilitate easier flow of the RF current [92]. The resulting selective RF current causes mechanical, electric, and molecular changes in the tissue [93,94,95].

An external field, ${E_{e}=10}^{4}\frac{V}{m}≅100\frac{mV}{cell}$, creates different SAR values in the intra- and extracellular electrolytes and in the membrane. The TME ${SAR}_{M+TME}>10^{8}\frac{W}{m^{3}}\approx 10^{5}\frac{W}{kg}\sim 1\frac{pW}{cell}$ [96]. The 10 MHz range has the highest absorption rate of the cell components [96]. The selection of the tumour by mEHT is therefore possible because of the macroscopic thermal and electric heterogeneity [93].

In silico calculations reveal [24,94] that the selection of the malignant cells using the electromagnetic processes produces a 2 °C higher temperature in the deep-seated tumour target than in the surrounding tissues [97]. Chopped meat phantoms, used to imitate tumour tissue, also showed a selective increase in temperature [98].

Experiments indicate that the cellular effects of treatment with mEHT are more enhanced than those seen with wHT at the same temperature [28,99] and in vivo [100,101]. This difference is likely driven by the synergy of the thermal and non-thermal effects.

In vivo studies on murine models show that mEHT can selectively target tumours, even under challenging conditions, such as lung metastases where the heating is complicated by the air flow [101,102]. The SAR in tumours reaches 250 W/kg, while in the surrounding tissues it is 125 W/kg [94], demonstrating the macro selection by the electromagnetic field with mEHT. The macroscopic thermal effect of mEHT also makes precise tumour temperature mapping possible [94].

The energy dissipation of RF in materials is proportional to the conductivity ($\sigma \left[\frac{1}{\Omega \cdot m}\right]$) of the absorbing material and the square of the electric field strength vector ($E^{2}$). There are two calculations which can be used to determine the SAR, or the power in the absorbing volume or mass:(6)$$SAR\;\left[\frac{W}{m^{3}}\right]=\sigma {\left|E\right|}^{2}\mathrm{or}\;SAR\;\left[\frac{W}{kg}\right]=\frac{\sigma }{\rho }{\left|E\right|}^{2}$$
where ρ is the mass density of the material. Furthermore, the current density (j) is calculated as follows:(7)$$j=\sigma E\left(\frac{A}{m^{2}}\right)\;$$

Consequently,
(8)$$SAR=\sigma {\left|E\right|}^{2}=\frac{1}{\sigma }j^{2}\;\left[\frac{W}{m^{3}}\right]$$

Under homogenous conditions, the measurement of $j^{2}$ offers a thermal dose in the material with σ conductivity. The thermal dose could therefore be described as follows: ($dose=\frac{energy}{mass})=\int SARdt=\int \frac{j^{2}}{m\sigma }dt$, [$\frac{J}{kg}$]). This simple dose monitoring equation could be applied if current matching is highly efficient and the energy loss through cooling mechanisms is minimal [18], as is the case with mEHT.

Temperature measurement is mandatory in radiative heating processes where non-selective heating is isothermal and the absorbed energy cannot be accurately calculated. The electromagnetic macro selection of tumours allows for the quantitative evaluation of the equivalent radiation dose [103], describing the synergy between radiotherapy and mEHT. This synergy has been shown in two different lung cancer cell lines, A549 and NCI-H1299, and the enhancement of the equivalent radiation dose by mEHT was confirmed in mouse xenograft models [103].

It has been shown that the temperature increases with an approximately linear growth rate and without disturbing the thermal homeostatic feedback in the beginning of the treatment [104]. The SAR, in this case, is proportional to the temperature development in time (t): $SAR≅c\frac{dT}{dt}$.

With precise impedance matching [27], the appropriate current density (j) is achieved. In an ideal conductor, the j current density and the E field immediately follow each other, while during the polarization process, a time lag (phase difference) appears between them. The electric current j acts thermally (Joule heat), expressed in SAR, and so the absorbed thermal power density depends on $j^{2}$ (8). The non-thermal effects, such as molecular excitations, polarization, structural changes, etc., proportionally depend on the electric field E(7). E linearly drives the non-thermal processes with j, ($E=\frac{1}{\sigma }j$).

Impedance-matched selective HT uses the current density j as an isodose parameter. As shown above, j does not depend on the size of the applied capacitor plates when the potential is kept constant. It is noteworthy that j depends on the reverse portion with the radius of the electrode when the power is unchanged; so, the power must be adjusted according to the electrode radius.

The applied electric field strength vector (E) creates the $j=\sigma E\left(\frac{A}{m^{2}}\right)$ dose according to Equation (7). Consequently, the thermal and non-thermal effects represent the nonlinear (${\sim j}^{2}$) and linear (~j) dependences of the selective dose and the energy absorption of cancer cells. The nonlinear thermal effect ensures the general energy background, while the non-thermal effect may be resonant [105].

When j is as large as the thermal effect dominating the interactions, it renders the linear non-thermal part negligible and produces conventional homogeneous heating without selective effects. The selection of the malignant cells, without heating healthy host cells, requires a lower j, where $j>j^{2}$, and subsequently a relatively small $\left(j\leq 1\right)$ current density. The thermal and non-thermal effects are equal when j=1. This happens when the membrane rafts, the TME, and the target tissue have the same temperature and no selection, the heating becomes homogeneous, and the thermal effect dominates the complete cellular process.

The incident power determines the thermal dose by $j\sim \sqrt{P}$. Heating excites the selected molecular clusters and promotes essential immune-related processes. Maintaining the temperature compensates for the energy losses, requiring a lower dose. The unchanged temperature with a lower current density produces less apoptosis compared to the active heating periods [106]. This offers the opportunity to improve the heterogenic cell destruction using mEHT. Most of the heating processes follow a step-up heating protocol [107], gradually increasing the dose to maximize apoptosis. Apoptotic cellular degradation determines the treatment’s efficacy [106]. The apoptotic processes linearly depend on the current density dose j of mEHT [108].

The absorbed thermal power (SAR) heats the membrane rafts, with the proteins absorbing more energy from the RF current that the lipid membrane [24]. The 13.56 MHz frequency directly targets the membrane lipid rafts [22], which appears as nano-heating because the absorbers, the membrane rafts, are nanoscopic.

### 5.2. Micro Selection

In silico models show that the induced electric field is highest in the cell membrane, which is constantly $\approx 5\cdot 10^{3}\;MHz$ times higher than the extracellular field, and the ratio between the two decreases by a power of 1/f as the waves approach higher frequencies. In the case of more realistic tissue models, the membrane’s gain depends on the cell’s position in the tissue, but it does not drop below $10^{2}$ in tissue arrangements [109] and it is double that of their healthy counterparts [109]. The voltage drop at the membrane excites the raft proteins [110], triggering various processes [111]. Experiments with gold nanoparticles demonstrated mEHT’s micro selection, showing higher energy absorption but decreased apoptosis due to energy sharing between membrane rafts and nanoparticles [112]. The energy-sharing phenomenon supports the theory of micro selection by mEHT.

During cytokinesis, the “neck” between daughter cells in the cell cycle’s telophase→cytokinesis induces cataphoretic forces [22] and absorbs a high SAR depending on its directional position to the E field, potentially arresting cytokinesis [113]. While the temperature increases at the points of contact, there is also a significant non-thermal effect seen during cytokinesis involving the cellular connections. The tumour treating field (TTF) is a widely applied field that focuses on the non-thermal effects of fields on the cytokinetic “neck” using capacitive coupling [114]. The electric field of TTF reorients microtubules and actin fibres, inhibiting mitotic spindle assembly [115,116]. The TTF is exclusively sensitive to micro selection due to its low external power, which allows it to prevail without any dominant macroscopic (heating) effects. The electric field increases the membrane permeability [117], which promotes the uptake of chemotherapy; however, at low frequencies, the double lipid membrane completely isolates the cytosol [118].

While the TTF applies a low external power, mEHT balances the low-power thermal effects and non-thermal electric processes. Modulated electro-hyperthermia uses a higher power input than the TTF, but still lower than conventional HT [114]. The mEHT treatments have $\sim 7.5\cdot 10^{4}\;V/m=75\;mV/\mu m$ field strength effect on the rafts [24] (and the necks), which is in the range of the membrane potential, and so its impact is much higher than ~2.2 mV/μm [119].

The patient appears here as a part of a tuned electric circuit, and the target’s impedance regulates the adaptation process. The low impedance of the tumour (macroscopic) and the TME (microscopic) drives the current to select the malignant cells. In this way, the mEHT achieves precise, in situ, real-time control of the intratumoural processes whilst accounting for the patient’s physiology [120].

### 5.3. Nano Selection

#### 5.3.1. Electromagnetic Processes


(a)Frequency Selection


Macro and micro selection target the ECM and TME, respectively. Using an RF current, the characteristic selection can be optimised [121], with the ideal range being in the MHz range [96,100]. In the context of dielectric spectroscopy and the study of biological tissues, dispersion ranges refer to different frequency ranges in which the dielectric properties of tissues change due to various relaxation mechanisms. Defining the dispersion ranges varies in the literature and this overlap is due to the wide distribution of effects that define them and their strong dependence on the target material.

Maxwell–Wagner relaxation [122] is associated with β dispersion focuses on the polarisation of cell membranes and the interfaces between different dielectric materials, such as cell membranes and the ECM [123]. This process involves the relaxation of dipoles in membrane-bound water and other components, which therefore modifies the membrane–TME interfaces, altering polarization and rearranging the charge distribution. The broad range of β-frequency dispersion effects [105] enhances the selective treatment by targeting these cellular components effectively [124]. Modulated electro-hyperthermia uses a 13.56 MHz frequency, which belongs to the medically applicable ISM band [125]. This frequency is generally associated with the second Maxwell–Wagner relaxation range, the δ-dispersion range [126], and interacts with TME-suspended particles [127], influencing organelles and protein-bound water [128].

The theoretical [96,129], and practical benefits (including antibiotic effects [130,131]) of both the β/δ dispersion ranges are well documented. Model calculations [132] show the 13.56 MHz frequency’s role in these dispersion regions, leveraging the recognizable impedance characteristics [133] that impact the TME, ECM, and cancer cell membranes [134]. The β/δ dispersion ranges select malignant cells, promoting energy absorption in membrane rafts, which influence transport and signal transduction [135], are involved in numerous signalling pathways, can sense various stresses [136], and modify the response external electromagnetic fields [137]. The application in this range can therefore enhance the selection process, provided that the power output and thermal effect do not override the non-thermal effects [138]. The 13.56 MHz frequency used by mEHT promotes non-thermal interactions, targeting transmembrane proteins and the water-bound lipid–protein complexes in the rafts. Rafts, sized 25–700 nm [139], promote the interaction of the transmembrane proteins [140] and trigger intracellular signals [111], and appear to be the primary energy absorbers in the target volume, making them a potentially important focus in cancer therapies [141].


(b)Modulation


To support healthy processes, homeostatic control manages transmembrane protein excitation, promoting natural cell degradation. Large transmembrane proteins are less likely to vibrate at high frequencies, and therefore require lower α dispersion frequencies (10 Hz<f<10 kHz) [142]. However, these lower dispersion frequencies can also stimulate muscles [143], which may be undesirable in certain contexts. To specifically target large molecules without undesirably stimulating muscle fibres and neurons, a high frequency is applied with modulation, creating an effective frequency within the α dispersion range [25]. This modulation leverages the selective advantages of β/δ dispersion frequencies for molecular excitation, enhancing the non-thermal components of absorption and promoting malignant cell degradation [105]. Experimental evidence supports the effectiveness of such modulation [25,26].

Modulation within the α dispersion range may enhance tumour-specific energy absorption [11]. Cell membranes rectify the signal [144,145,146], converting the alternating current to a direct current and resulting in the accumulation of signal from the external alternating current electric field. The rectified potential level is approximated to 1 μV, which does not directly modify the membrane potential but could induce significant ion fluxes and cell disequilibrium. The resistivity and capacity of the TME and cell membrane determine the time constant of the detection of stimuli and changes in the environment [11].

Autocorrelation, a statistical tool that helps assess how current signals relate to past signals in the same pathway, can be used to analyse the regularity and predictability of signals within these homeostatic pathways. Consistent patterns (i.e., positive autocorrelation) can indicate normal, healthy functioning [147,148], and the mixture of all signals in a healthy system forms a structured pattern of fluctuation, referred to as “noise” [149]. This noise is structured and maintained by homeostasis. The noise density *S*(*f*), or power at a certain frequency *f*, decreases as the frequency increases, as described by the equation
(9)$$S\left(f\right)=f^{-\alpha }$$

This relationship holds true at all levels, from molecules to the entire organism [150], and the noise density pattern is most evident at low frequencies. External signals could therefore be applied to enhance signals from these healthy pathways [151], selecting chemical reactions based on their timing and sequence to promote healthy enzymatic processes.

The α parameter in the healthy system is α≅1, and in this state, the relative error of cellular fission remains low. When the system α deviates from α≅1, indicating a problem in homeostatic control, the errors in cellular processes such as cellular fission increase [152]. In this way, applying Function (9) can help to differentiate between healthy and malignant cells [153,154] and the principles of chaos theory can be used to understand and control biological processes [155,156], as a healthy system will show ordered chaos [157].

Modulated electro-hyperthermia applies the signal mixture equivalent, $S(f)=f^{-1}$, as an amplitude modulation of the 13.56 MHz carrier, much in the same way frequencies are modulated to deliver radio broadcasts (audio frequency, α dispersion range) to the listeners.

#### 5.3.2. Molecular Selection

A cylindrically shaped lipid raft with an average diameter of 100 nm and a thickness of 7 nm has a volume of ${5.5\cdot 10}^{-23}m^{3}$ and absorbs extremely large amounts of energy calculated in the following cubic meter raft volume: $SAR\approx {2\cdot 10}^{9}\frac{W}{m^{3}}$. This extreme SAR heats only a thin volume, affecting only the raft with a low power: ${SAR}_{raft}\approx 0.11\frac{pW}{raft}$.

Models show that this SAR heats the raft to around ~4 °C higher than its environment [24]. Comparative studies of wHT and mEHT in various cell lines demonstrate mEHT’s thermal and non-thermal effects [18,22,28,31,45,55]. Rafts are highly heterogeneous and dynamic cholesterol-enriched micro-domains that participate in protein interactions and compartmentalize cellular processes [158], promoting regulatory molecule binding [159]. Rafts also trigger apoptotic signalling, such as tumour necrosis factor (TNF)-related apoptosis-inducing ligand (TRAIL) death receptors, which are integral to immune surveillance. Non-thermal effects from mEHT excite transmembrane proteins in rafts, inducing molecular and structural changes [159,160].

Malignant cells have a denser raft population than non-malignant cells [161], enhancing their selection by mEHT. The high number of membrane rafts in malignant cells absorbs significant energy [24], leading to changes in nano-mechanical properties [162], such as increased membrane compartment motility and rigidity [163]. The external electric field stabilizes the membrane raft domains in the lipid bilayers [164,165]. This, as well as the relatively low membrane potential of cancer cells [166,167], further enhances the mechanisms for nano selection.

#### 5.3.3. Global Gene and Protein Expression Profiles

The micro selection of mEHT also appears at the DNA level, involving DNA fragmentation [168,169]. Synergistic thermal and non-thermal effects from mEHT impact DNA repair, suppress the Ki67 proliferation marker [170], and suggest a connection between DNA double-strand breaks and mEHT treatment. Gene regulation differences between wHT and mEHT are observed in various cell lines and in vivo models [108,171,172]. The gene map shows a distinct difference in gene regulation between the homogeneous wHT and inhomogeneous mEHT treatments [108] at the same 42 °C temperature. For example, a gene analysis of human lymphoma cell lines (U937) heated to 42 °C with wHT showed stimulation of the cytoprotective gene network. The HSP105 and HSP90A genes were upregulated, inhibiting caspase activation and blocking apoptosis. In contrast, mEHT at the same temperature activated anti-tumoural activity in HSPs, with the activation of specific genes for cell death, such as EGR1, JUN, and CDKN1A. Modulated electro-hyperthermia did not activate cytoprotective pathways like wHT, instead dominantly activating the FAS, JNK, and ERK signalling pathways [108], suggesting that electromagnetic effects influenced the differences in cell fate. The special characteristics of the electromagnetic field [173] on genes in the case of mEHT force apoptosis in vivo xenografts of hepatocellular carcinoma [174], gliomas [172], colorectal [171], and isograft tumours of triple-negative breast cancer in preclinical models [170].

The upregulated genes in 4T1 triple-negative breast cancer isografts show significant alterations in stress response pathways [170]; the humoral immune response (21 genes, $p=1.38\cdot 10^{-5}$); serine-type endopeptidase inhibitor activity (10 genes, $p=1.9\cdot 10^{-5}$); complement binding (8 genes, $p=2.41\cdot 10^{-4}$) and extracellular matrix structural constituent binding (39 genes, $p=4.9\cdot 10^{-11}$) are all impressive indicators of changes. The 38 upregulated genes ($p=1.2\cdot 10^{-4}$) in the “response to stimulus” pathway also vary highly significantly.

A transcriptomic analysis of mEHT-treated cells revealed a genetic upregulation of E2F1 and CPSF2 and the downregulation of ADAR1 and PSAT1, leading to strong anti-proliferative effects on cancer stem cells (CSCs) [172]. The increases in E2F1 and p53 levels and the decrease in the PARP-1 level caused an enhanced apoptotic signal in U87-MG and A172 cells. mEHT treatment caused a drastic reduction in the number of CD133 CSCs and suppressed malignant cell migration [172]. Lastly, the combination of the thermal effect and the ionizing radiation field inhibits the proliferation of cancer cells, involving the NR4A3 and KLF11 genes [175].

## 6. Effects of Non-Thermal Selection

### 6.1. Enzymatic Effects

The external electric field interacts with transmembrane proteins and triggers ionic effects in the TME [105], influencing macro processes [176] and catalysing reactions that are frequency dependent [177]. Enzymes, which catalyse biological processes by lowering activation energy barriers, can be affected by modulation, enhancing enzymatic processes [178,179]. The modulated electric field impacts enzymatic processes [180] through electro-conformal coupling (ECC, [181,182]), which has a resonant behaviour [183]. The low-frequency amplitude-modulated carrier frequencies generate stochastic resonances (SRs), inducing various biological enzymatic reactions and polymerization processes [105], maintaining homeostatic equilibrium [184], and optimising energetic processes [185,186].

Stochastic resonance (SR) happens when a periodic signal is mixed with noise, amplifying weak signals. Saturation occurs with increasing frequency or noise intensity, necessitating optimization for maximal SR [187]. The actual resonance depends on the environmental temperature, producing conditional noise. In a healthy system, cells form a “democracy” [188], exchanging information and aiding tissue regeneration and emitting healthy 1/*f* pink noise [189]. Cancer cells, however, prefer autonomy, disrupting multicellularity [190,191,192]. In this way, the applied modulated field can affect enzymatic pathways, working against malignant processes.

### 6.2. Stress Proteins

Thermal stress induces protective chaperoning heat shock proteins (HSPs) [193], which prevent protein aggregation and denaturation and correct the structural damage of the proteins resulting from stress [194]. HSPs, which are expressed under various stresses, can inhibit apoptosis, promote metastasis, and influence immune responses [195], and these effects help to protect the cell. Malignant cells, which have high baseline levels of intracellular HSPs (iHSPs), use the iHSPs to adapt to stress and to survive [196].

While healthy cells develop a relatively large number of iHSPs after thermal stress (an increase in 8–10 times the normal number at 42 °C), malignant cells only increase the number iHSPs by 30–40% [197]. Healthy cells may share the absorbed energy with their neighbours through the cellular network, but malignant cells absorb it individually, making them more vulnerable to heat.

The non-thermal electromagnetic stresses also induce chaperone synthesis [198], potentially enhancing the immune response [199] and promoting the formation of intercellular bonds [200]. Modulated electro-hyperthermia increases HSP expression at mRNA [171] and protein [201] levels, as observed in murine models.

iHSP70s protect the cancer cells from apoptosis and improve their resistance to further stresses [193] but may promote cell death when dysfunctional [202]. The dual role of HSPs in malignant cells as protective or destructive agents [203,204] depends on the intra- and extracellular conditions [205].

The HSPs liberated into the ECM (eHSPs) may significantly promote anti-tumoural activity [199]. The iHSPs released into the TME have a complex translocation process, including lipid interactions [206]. During the translocation process, an intermediate state [207] is noted, during which time the HSPs are membrane bound, forming membrane HSPs (mHSP) [208]. The mHSPs induce apoptosis [200] and, depending on the conditions, either support [209] or suppress the survival [210] of the malignant cells. mHSPs attract NK cells [211] and activate innate immune surveillance. The extracellularly released HSPs (eHSPs) deliver immunogenic information to APCs for tumour immunity. Necrotic processes also may provide eHSPs without the iHSP→mHSP→eHSP conversion.

### 6.3. Ion Channels in the Cell Membrane

Modulated electro-hyperthermia affects membrane potential fluctuation, which is crucial for ion transmission [212]. The electric field impacts ion transport [213], cellular signalling [214], the transmembrane potential [215,216], and DNA transfection [217,218], potentially influencing the immune system through membrane-mediated calcium signalling [219]. The synergy of thermal and non-thermal effects modifies intracellular molecular motors and ion-exchanging channels [186,212,220]. The electric field activates ion transport, such as sodium–potassium ATPase [221], and increases the intracellular calcium (*iCa*^2+^) concentration [11,222]. External RF exposure has an impact on erythrocyte haemolysis [223] and may activate the ion channels in the cell membrane [224,225]. The electric field may influence the immune system through membrane-mediated ${Ca}^{2+}$ signalling [219] and alter the transmembrane ${Ca}^{2+}$ at low-frequency electromagnetic fields [226].

A comparison of wHT and mEHT under the same thermal conditions also shows significant differences in the intercellular calcium ($i{Ca}^{2+}$) concentration [28,30], [99,108,146]. An overload of ${iCa}^{2+}$ is detrimental to the cell and may cause apoptosis. The modulated field from mEHT therefore also appears to cause an increase in the ${iCa}^{2+}$ concentration, which likely contributes to the destructive effects of mEHT on malignant cells [11].

### 6.4. Transient Receptor Potentials (TRPs)

Temperature-sensing TRP proteins in the cell membrane detect thermal changes. Four TRP subtypes are heat-sensitive [111,227], and several subtypes regulate calcium transport through the membrane. The rafts contain TRP channels, and so when applying mEHT, the selection of the rafts may directly activate TRP regulation. Stimulation of the TRPs [99] increases ${Ca}^{2+}$ influx to critically high levels [99], and the high $i{Ca}^{2+}$ concentration promotes apoptosis through the mitochondria-dependent intrinsic signalling pathway [228]. TRPV4 is a ${Ca}^{2+}$-permeable channel that works at body temperature. The temperature modulates the activation of a diverse range of microenvironmental chemical and physical signals, functioning as molecular integrators [229]. The integrative behaviour of TRPV4 is therefore active in the presence of moderate (under 40 °C) HT processes, as it senses changes in the TME. The combination of thermal and non-thermal processes creates a complex synergy, with TRPV4 being a key target for cancer treatment. Additionally, the thermal processes have a combined effect with non-thermal electric (voltage-gated processes) and chemical (capsaicin excitation) processes.

### 6.5. Effect on Polymerization of the Cytoskeleton

The cytoskeleton, composed of actin and tubulin polymers, frequently deviates from its normal structure in malignant cells, affecting cell shape and motility [230,231,232,233,234]. The external electric field can reorganize actin filaments and potentially block cytoskeletal reorganization, stopping cytokinesis [235] and blocking cellular fission. The modulated field may promote apoptotic signal transmission [236] by applying harmonizing noise [237]. Furthermore, voltage-sensitive phosphatases (VSPs) play a role in cytoskeletal restructuring [238], and these molecules may be influenced by the electric field to support correct network building.

Cytoskeletal fibres follow the external electric field [239] and may orient the galvanotaxis [240]. An alternating current (AC) reorganizes the cytoskeleton with an optimum resonant frequency of around 1 Hz [241]. Theoretically, stochastic resonance describes this phenomenon [105]. The cytoskeleton works like a vast and complex wiring network for signals. The pathways are continuous from start to finish, and so the polymer density must be over the percolation threshold, forming long-range connectivity in a random system [242]. A continuous giant component of the order of system size exists above the threshold, but no such connectivity exists in the system below it. The excitations and ionic species move in these giant pathways much quicker than the simple diffusion allows it [243].

### 6.6. Apoptotic Signals

Modulated signals can trigger apoptotic excitation of transmembrane proteins and signalling pathways [244]. The extrinsic apoptotic pathway is initiated at the TRAIL death receptor, involving FADD and FAS complexes [201]. TRAIL signalling can have both pro- and anti-tumour effects, depending on the TME [245]. The modulated signal directs TRAIL towards apoptotic processes through both the extrinsic and intrinsic pathways, and it activates the caspase-dependent extrinsic apoptotic pathway.

As previously described, the stress caused by the modulated field causes the overexpression of iHSP70 [246], which exhausts cell performance and facilitates apoptotic cell death [101]. Furthermore, the expressed iHSP60 proteins activate the cleaved form of Cas3 favouring caspase-dependent apoptosis [247].

Additionally, the activation of p21^waf1^ by mEHT, as observed in vitro in human Panc1 [168] and C26 mouse colorectal cell lines, in vivo in A2058 and B16F10 melanoma allografts [102,248], as well as their corresponding lung metastases [97], shows significant tumour growth inhibition. The rapid increase in the expression of the CDKN1A gene, encoding p21^waf1^, mediates the DNA double-strand break response triggered by p53 activation [102]. This results in substantial tumour growth suppression and cell cycle arrest during metastatic melanoma growth in mouse lungs [97].

The enhancement of apoptotic signals by mEHT works synergistically with ionizing radiation, enhancing the effects of radiation therapy on cancer treatment.

### 6.7. Chemotherapy-Boosting Field Effects

Low-frequency external electric fields may enhance the effectiveness of chemotherapy [249] and bypass multidrug resistance [250]. These fields have been shown to increase the sensitivity of cells to chemotherapies like paclitaxel and doxorubicin [251,252]. For instance, the penetration of doxorubicin improves when combined with 50 Hz electrical stimuli in drug-resistant cancer cell lines [250].

Phase III clinical trials on newly diagnosed glioblastoma multiforme patients [253] show that tumour treating fields (TTFs) combined with temozolomide chemotherapy significantly improve outcomes with negligible side effects. In trials for recurrent gliomas, TTFs results in comparable survival rates to various chemotherapies [254].

Additionally, low-frequency modulated electric fields promote liposomal drug uptake in cell lines such as HepG2, U87MG, A459, and CT26 [255]. Liposomal targeting enhances drug delivery to malignant cells while reducing uptake in healthy tissues.

The non-thermal electroporation by an electric field creates membrane permeabilization [256,257,258], enhancing drug penetration [259,260]. This technique is effectively combined with chemo-radiotherapy [261] and has been successfully applied in treating metastatic complications [262].

## 7. Immunogenic Effects

While the selective heating of mEHT delivers high temperatures to malignant cells, the average temperature of the tumour remains below 40 °C. This has an immunogenic advantage: temperatures above 40 °C downregulate the cytotoxicity of innate immune attacks [263], including those by natural killer (NK) cells [264]. Beyond thermal stress, non-thermal mechanisms of electric fields, aided by applied modulation, synergistically promote a specific set of molecules known as damage-associated molecular patterns (DAMPs).

The thermal and non-thermal effects of mEHT cause an increase in reactive oxygen species (ROS) levels that peaks 3 h after the treatment, as seen in HepG2 cell lines treated mEHT. Cells treated with mEHT exhibit relative fluorescence levels (4.87) three times higher than those treated with conventional HT (1.54) [32]. This activates the protein kinase R (PKR)-like endoplasmic reticulum kinase (PERK) and initiates the unfolded protein response (UPR), triggering the production of the endoplasmic reticulum (ER) chaperone BiP (HSP70 equivalent) [265] to restore cellular health. In efforts to re-establish the normal structures, the UPR aims to promote apoptosis and liberate eATP as a second-class DAMP [247].

Chaperone functions intensify throughout the cell, producing large amounts of iHSPs to protect the cell’s integrity, which is often pro-tumour. However, in vitro and in vivo measurements show that mEHT induces apoptosis without comparable necrosis in malignant cells. The iHSPs have potential to transition from protective to destructive roles as malignant cells secrete HSP70 partially on the cell membrane (mHSPs) and partially into the extracellular matrix (eHSPs) [244]. In HT29 colorectal carcinoma xenograft experiments, iHSP70 levels peaked at 14 h post-mEHT treatment, returned to baseline at 48 h, and showed a second peak between 72–120 h, due to mHSP and eHSP secretion [244]. Similar observations were made in mEHT-treated B16F10 melanoma allograft experiments, with increased mHSP70 and eHSP70 levels and substantial apoptosis at 48 h post-treatment [102]. When compared to other heating methods, the levels of eHSPs at 24–48 h post-treatment were significantly higher in the cells treated with mEHT [28]. The formation of iHSP60 by mEHT [201] promotes the cleavage of Cas3 [247], inducing caspase-dependent apoptosis.

In the early 1950s, Mole RH described the abscopal effect, where the local application of ionising radiation has the potential to affect lesions outside of the treated field [266]. This effect is now gaining significant interest [267,268] and is believed to be driven by immunogenic processes.

Thermal and non-thermal energy absorption by membrane rafts triggers signalling pathways, producing ICD and DAMP molecules. The non-thermal excitation by mEHT orchestrates the temporal and spatial appearance of these molecules, potentially enabling remote actions on distant metastatic locations and disseminated tumour cells [269]. Applied modulation ensures that the correct set of DAMPs appears in the appropriate order. This has been adequately described elsewhere in the literature [116].

The spatiotemporal DAMPs induced by mEHT guide ICD in an anti-tumour direction [265]. Most of the tumour cell destruction by mEHT promotes apoptotic cell death through the immunogenic pathways, producing a set of DAMPs [101,102,108], [171,196,201]. The immune system may recognize eHSP70 as an immune messenger transporting intracellular antigenic peptides to antigen-presenting cells (APCs). The genetic information carried by eHSPs drives DC maturation [270], producing active molecules (cytokines and chemokines), immune cells (NK cells and tumour-specific ${CD4}^{+}$ and ${CD8}^{+}$ and γδT-cells), and activating antitumour immunity [271,272].

The abscopal effect requires DCs to mature into APCs. Various pre-clinical studies have shed light on the immunogenic and abscopal effects of mEHt [97,101,273,274]. These processes determine the tumour-specific immune response, which can detect and eliminate micro- and macro-metastases throughout the body, as is seen during vaccination procedures (Figure 1).

## 8. Conclusions

The complex interplay of thermal and non-thermal effects makes mEHT a unique therapeutic approach. The synergy of these effects enables the precise targeting of malignant cells, with minimal impacts on healthy tissue. The method leverages the thermal and electrical heterogeneities of tissues. The applied radiofrequency (RF) signal is modulated to induce various apoptotic pathways, creating optimal conditions for non-thermal excitation processes. This combination is fine-tuned to induce immunogenic cell death (ICD), which triggers the secretion of damage-associated molecular patterns (DAMPs). DAMPs gather immunogenic information from the tumour, facilitating the presentation of tumour-specific antigens to T-cells, enabling them to recognize and target malignant cells throughout the body. This process extends the local effects of mEHT to a systemic level, allowing the simultaneous targeting of distant micro- and macro-metastases. The locally treated tumour provides real-time immunogenic information, supporting the abscopal effect. The formation of tumour-specific memory T-cells mirrors the mechanisms of vaccination, offering a promising approach to cancer treatment.

## Figures and Tables

**Figure 1 cancers-16-03908-f001:**
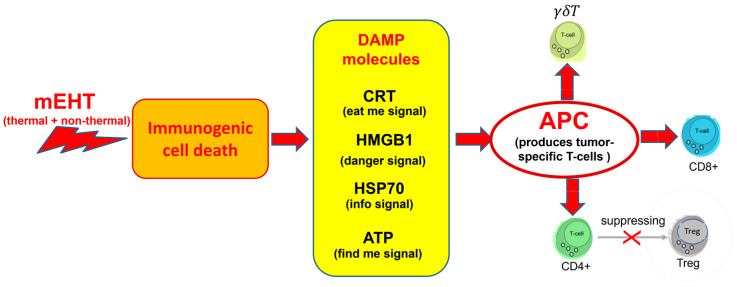
The combination of thermal and non-thermal effects creates immunogenic cell death, which enables the formation of a special molecular set in the TME. The appropriate molecular combination promotes the formation of antigen-presenting cells and thus enables the tumour-specific formation of killer and helper T-cells, suppressing the controversial Tregs.

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
