# Peer review of "The Synergy of Thermal and Non-Thermal Effects in Hyperthermic Oncology"

_cancers, 2024, doi:10.3390/cancers16233908_

Round 1
Reviewer 1 Report (Previous Reviewer 1)
Comments and Suggestions for Authors I reviewed the point to point response along with the revised manuscript.I think the manuscript has greatly improved and the authors responded well to all raised points. I therefore recommend to accept this manuscript in its current form.
Author Response
Dear Reviewer,
We would like to say thank you for your work and for recommending the manuscript for publication in its form.
Kind regards,
the Authors
This manuscript is a resubmission of an earlier submission. The following is a list of the peer review reports and author responses from that submission.
Round 1
Reviewer 1 Report
Comments and Suggestions for Authors
Dear Editor,
thank you for the opportunity to review this interesting paper. This topic is very important and relevant to the oncological community and therefore deserves to be published and will have impact. I think in its present form it is not easy to read. The current version to my understanding suggests that the way “thermal” and “non-thermal” effects condition oneself is clear, but I fear this is actually not the case. Providing to much detail information and too many references is not improving this and I therefore believe the message and flow of the manuscript could be improved when some revisions are performed. This will also increase the citations of the published manuscript when published later in CANCERS:
Major remarks:
1) Simple summary, abstract and manuscript: I wonder why the authors did not introduce capacitive hyperthermia and made the point that this comes already with potential nonthermal effects. Then it would be an important next step to show that additional modulation (mEHT) has the potential to further improve these nonthermal effects.
2) Please doublecheck the use of abbreviations in the manuscript draft. Often introduction of abbreviations is too late and later not used consistently. This kind of distracts the reader (and reviewer) from the important content of the manuscript.
Examples:
Line 41 and 44: hyperthermia
Line 59 and 65: Local-regional hyperthermia vs. LRH vs. Local-Regional hyperthermia
Line 282: mEHT is introduced on page 6 and again line 335 on page 8 and again on line 515 on page 11
Line 343: wHT? Introduced on line 620 and again line 667 page 14 but used before line 618
3) Introduction: All the effectiveness data cited comes from RF-based hyperthermia. Why WBH that comes with hardly any solid data in cancer patients is required here? Should be made clear at least that there is no supportive outcome data for WBH and this actually supports the hypothesis of the authors (compare also lines 75-78)
4) Lines 80 – 104 are problematic.
To mention WBH (locally) here again distracts from the point that the regional RF based approaches were successful in several randomized trials.
To mix mEHT with nanotherapy in the next paragraph is even more troublesome.
I appreciate that there is a hypothesis with lipid rafts that was previously described several times but I think it would make more sense to discriminate capacitive hyperthermia from radiative hyperthermia devices and make the point that mEHT goes beyond this for several reasons (to be pointed out).
I believe in the current version the information can not be understood by the average people working in hyperthermia which is problematic. This is in part because the nomenclature that is established (compare above) is not used.
Regarding line 92-104: First paragraph: Well MR-guided thermometry and thermometry in general could at least in part account for this. Second paragraph: I do not get why you write here about the disadvantages of nanoparticle administration (you have not even introduced the hypothesis with the lipid rafts which comes only in the subsequent paragraph)
5) Lines 106-118: Please make clear whether lipid rafts are a target for pure sinusoidal RF (like thermothron) or only for an amplitude modulated signal such as mEHT. I would bring this information earlier and make it more clear. Some other parts of the long introduction could be left out.
6) Lines 147-162: Where does the f > 15 MHz come from? Tumor treating fields (Novocure) are using > 1kHz with the argument that the membrane becomes conductive. This is probably very much the case at 13.56 MHz, or not? (Elson E. Biologic effects of radiofrequency and microwave fields: in vivo and in vitro experimental results. In: Bronzino JD, editor. The biomedical engineering handbook.Boca Raton, FL: CRC Press, Inc.; 1995. p. 1417–23.)
7) Lines 164-172: Is this a theory, if yes please make this clear by using words such as “may”
8) Lines 190-192: Where do you know that mitochondria, ER and Golgi apparatus and so on are impacted upon?
9) Lines 201-239: I suggest to focus on cancer as suggested in the title.
10) Lines 236-239: I have some problems with the phrasing here. Is this not very speculative??
11) Lines 240-241: How? Is it an nonthermal effect that is used or does mEHT cause this effect and how?
12) Page 7: Figure 1 appears hard to read
13) Page 7-10 and 12-20 are hard to understand for clinicians. The two clinician authors should please shorten this up.
14) Lines 498-510: These are all very valid points who have been stated elsewhere and contributed that TTF is only used in 5-10 % of patients with glioblastoma today. Why is this information important here??
15) Not clear why Theranostic line 673
16) Lines 845 on: Maybe I missed it but where is the information regarding voltage-gated channels?
17) Page 22: are this not predominantly heat effects??
18) Conclusion line 1139: Why not call it RF effects?
19) Line 1144: What does “is tuned” mean in this context?
20) I strongly advise to shorten the manuscript especially because otherwise the first part of the paper from pages 7-20 is hard to follow and the remaining part will not be read. I strongly advise to significantly reduce the numbers of references as well to <150-200
Minor remarks:
1) Not clear why abstract introduces abbreviations that are not used again in the abstract
2) Please shorten the introduction. No reason I believe to start with the ancient citation again…
3) Typo “ae” line 48
4) Typo “after the” line 192
5) Check the sentence starting at line 201
6) Line 310 “cells” instead of host??
7) Typo: multiform line 1009
8) Typos on page 21: at50, ROS 3h, ofHEP
Reviewer 2 Report
Comments and Suggestions for Authors
Observations:
I consider this research work to be very well structured, very complete and provides the necessary information to understand the challenges facing the development of Modulated Electro-hyperthermia (mEHT) in cancer treatment.
This manuscript makes it clear that from the thermal and non-thermal effects present in mEHT, advantages and disadvantages arise, which must be considered for its use in cancer therapeutics.
The authors address the proposed molecular mechanisms in mEHT-induced apoptosis, as well as Immunogenic cell death (ICD) and the involvement of DMAPS and dendritic cells.
I suggest looking at immunotherapies in oncology such as adoptive tumor-infiltrating lymphocyte immunotherapy (TIL) and CAR-T therapy compared to mEHT
I suggest placing more emphasis on the fact that death by necrosis, in addition to apoptosis, has been reported by mEHT. These are two molecularly different death phenomena. It would probably be worth emphasizing that there are different types of cell death and that the type of death seen in mEHT depends on different conditions of the tumor cell, as well as the treatment that has been given. Suggest that other types of death induced by mEHT need to be explored
In line 48 "ae" appears, I suppose the author meant to write "a"
In lines 101 and 102 it appears "It may be necessary to clean these organs of particles after the" I think the sentence is incomplete
Check line 364 (28Error! Marker not defined)
Check line 555 (Error! Marker not defined)
Check line 621 (Error! Marker not defined)
On line 1032 it says "ofHepG2" I guess it should say "of HepG2"
Check line 1121 (Error! Marker not defined)
